# In-Context Reinforcement Learning Without Optimal Action Labels

**Juncheng Dong** [1]  **Moyang Guo** [1]  **Ethan X. Fang** [2]  **Zhuoran Yang** [3]  **Vahid Tarokh** [1]

## Abstract

Large language models (LLMs) have achieved remarkable empirical successes, largely due to their in-context learning capabilities. Inspired by this, we explore training an autoregressive transformer for in-context Reinforcement Learning (RL). In this setting, we initially train a transformer on an offline dataset consisting of trajectories collected from various RL instances, and then fix and use this transformer to create an action policy for new RL instances. We consider the setting where the offline dataset contains trajectories sampled from suboptimal behavioral policies. In this case, standard autoregressive training corresponds to imitation learning and results in suboptimal performance. To address this, we propose the Decision Importance Transformer (DIT), which emulates the actor-critic algorithm in an in-context manner. DIT trains a transformer-based policy using a weighted maximum likelihood estimation (WMLE) loss, where the weights are based on the observed rewards and act as importance sampling ratios, guiding the suboptimal policy toward the optimal policy. We conduct extensive experiments to test the performance of DIT on both bandit and Markov Decision Process problems. Our results show that DIT achieves superior performance, particularly when the pretraining dataset contains suboptimal action labels.

## 1. Introduction

Large Language Models (LLMs) have achieved remarkable empirical successes (Radford et al., 2019; Brown et al., 2020b; Touvron et al., 2023; Wu et al., 2023b; OpenAI et al., 2024). In particular, colossal transformer models trained with copious data have shown astonishing in-context learning (ICL) capabilities (Akyürek et al., 2022; Dong et al., 2022; Min et al., 2022), i.e., to solve new tasks with only a few demonstrations (Brown et al., 2020a; Perez et al., 2021; Alayrac et al., 2022). In the setting of ICL for supervised learning, when presented with the context of a small batch of paired inputs and labels from a new task, LLMs generate the associated label for an unpaired input. This process does not involve any parameter updates; instead, LLMs rely solely on the provided demonstrations to determine the correct label. ICL has been successfully applied to a broad range of supervised learning tasks (Brown et al., 2020b; Xie et al., 2021; Min et al., 2022; Touvron et al., 2023). On the other hand, Reinforcement Learning (RL) represents a significantly more complex and greater challenge than supervised learning (Kaelbling et al., 1996; François-Lavet et al., 2018; Levine et al., 2020). In the online setting, RL algorithms constantly balance exploration and exploitation: choosing exploratory actions to gather more information at the cost of high regret, or exploiting the currently known optimal action with the risk of overlooking a potentially better one. In this case, RL algorithms act optimistically and explore with audacity (Garivier & Moulines, 2011; Kaufmann et al., 2012). Conversely, in the offline setting where all information comes from datasets collected by suboptimal behavioral policies, RL algorithms act pessimistically (Kidambi et al., 2020; Kumar et al., 2020; Rashidinejad et al., 2021; Jin et al., 2021).

Despite these challenges, recent works have successfully employed autoregressive LLMs for in-context RL (Laskin et al., 2022; Lee et al., 2024). In this setting, the context is an offline dataset consisting of transitions collected by unknown and often suboptimal policies. When used as policies, LLMs predict the optimal actions for current states conditioning on the given context. In this case, LLMs infer optimal policies from the environmental information provided in the context. Two recent works, Algorithm Distillation (AD) (Laskin et al., 2022) and Decision Pretrained Transformer (DPT) (Lee et al., 2024), have demonstrated impressive in-context RL abilities. After pretraining on datasets from a family of diverse RL instances, they generalize to *unseen* RL instances and infer near-optimal policies in context. As a pioneering work, AD adopts a learning-to-learn (Vilalta & Drissi, 2002) approach, training a transformer model to

---
*Equal contribution  [1]Department of Electrical and Computer Engineering, Duke University. [2]Department of Biostatistics and Bioinformatics, Duke University. [3]Department of Statistics and Data Science, Yale University.. Correspondence to: Juncheng Dong <juncheng.dong@duke.edu>.

*Proceedings of the $1^{st}$ Workshop on In-Context Learning at the $41^{st}$ International Conference on Machine Learning*, Vienna, Austria. 2024. Copyright 2024 by the author(s).

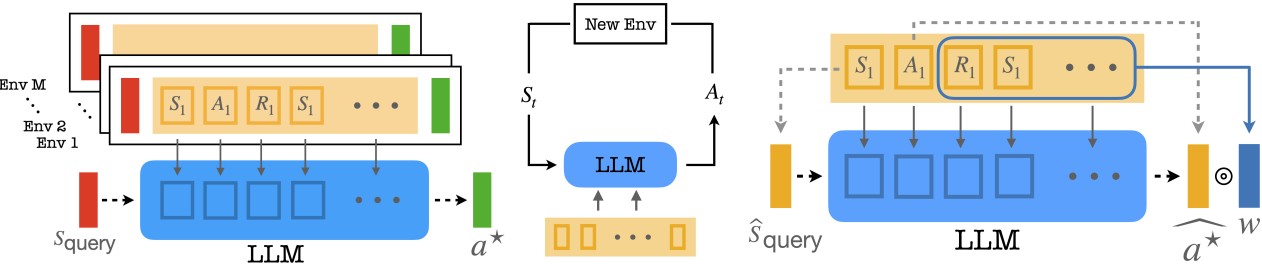

*Figure 1.* **Supervised Pretraining (left)**: Presented with offline trajectories and optimal action labels, LLMs are pretrained to predict the optimal actions for query states across RL tasks. **In-Context RL (middle)**: When used as policies for unseen environments, the pretrained LLMs generate actions conditioned on the current states and offline trajectories collected by (suboptimal) behavioral policies. **Pretraining without Optimal Action Labels (right):** This work addresses the stringent requirement for optimal action labels by using in-trajectory state-action pairs as query states and *pseudo*-optimal action labels. Additionally, it employs a *weighted* pretraining objective, where the weights are based on the optimality of actions, estimated through the observed rewards.

emulate the learning process of RL algorithms. In particular, AD requires the pretraining dataset to include the complete learning history of RL algorithms—from episodes collected by randomly initialized policies to those collected by nearly optimal policies—across a diverse set of RL tasks. This imposes a stringent demand on the pretraining dataset.

Our work is more closely related to DPT, a supervised pretraining method. The pretraining dataset for DPT consists of offline trajectories (contexts) from a diverse set of RL tasks, randomly sampled query states from these tasks, and their corresponding optimal actions. DPT trains the transformer to predict the optimal action given a query state and the context (see Section 3 and Figure 1). Despite its remarkable in-context RL ability, DPT requires access to optimal policies to generate the optimal action labels for a set of query states across diverse tasks. Although this requirement is less demanding than that of AD, it remains a stringent assumption. To address this limitation, we propose the *Decision Importance Transformer* (DIT), a framework to pretrain transformers for in-context RL *without* optimal action labels. DIT is based on a simple principle: *in the absence of query states and optimal action labels, can we use the observed state-action pairs in the offline dataset as query states and pseudo-optimal action labels?* Specifically, given offline datasets from diverse RL tasks, DIT assigns a value to each observed action in the datasets. The assigned values are based on the observed rewards in trajectory, representing the optimality of actions. The observed actions, along with their assigned values, serve as noisy estimations of the true optimal actions. A transformer is subsequently trained with a weighted maximum likelihood estimation (WMLE) loss, where the weights are based on the assigned values. These weights essentially act as the importance sampling ratio between the optimal trajectories and the offline trajectories, selecting the in-trajectory state-action pairs that can be reliably used as query states and optimal action labels

for supervised pretraining.

Through thorough experiments on various bandit and Markov Decision Process (MDP) problems, we demonstrate that pretrained DIT models also generalize to unseen decision making problems in context. On bandit problems, the performance of DIT models matches that of the theoretically optimal bandit algorithms (Upper Confidence Bound (Auer, 2002) and Thompson Sampling (Russo et al., 2018) for online learning; Lower Confidence Bound (Xiao et al., 2021) for offline learning). In the challenging MDP problem Dark Room (Laskin et al., 2022), DIT models demonstrate competitive performance to that of the DPT models in both online and offline testings, despite being pretrained without optimal action labels. To corroborate that DIT has extracted the maximum information from the pretraining dataset, we conduct experiments on Miniworld (Chevalier-Boisvert et al., 2023) where the query states for DPT are only allowed to be sampled from the observed states. In this case, even pretrained with optimal action labels, the performance of DPT models is only on par with that of DIT models.

In what follows, we briefly discuss related work on in-context decision making in Section 2, followed by a preliminary review of MDP and DPT in Section 3. In Sections 4, we present DIT for bandit and MDP problems. Experimental results are highlighted in Section 5, followed by our conclusions.

## 2. Related Work

**Offline Reinforcement Learning.** While online RL algorithms (Kaelbling et al., 1996; François-Lavet et al., 2018) learn optimal policies by interacting with the environments through trial and error, offline RL (Levine et al., 2020; Matsushima et al., 2020; Prudencio et al., 2023) aims to infer optimal policies from historical data collected by (suboptimal) behavioral policies. One of the substantial challenges for

offline RL is the distribution shift caused by the mismatch between behavioral policies and optimal policies (Levine et al., 2020; Kostrikov et al., 2021). To this end, offline RL algorithms learn pessimistically by either policy regularization or underestimating the policy returns (Wu et al., 2019; Kidambi et al., 2020; Kumar et al., 2020; Rashidinejad et al., 2021; Yin & Wang, 2021; Jin et al., 2021; Dong et al., 2023). While the goal of offline RL is solve the *same* RL tasks from where the offline datasets are collected, the goal of in-context RL is to efficiently generalize to *unseen* tasks after pretraining with offline datasets from diverse RL tasks. In particular, models with in-context RL capabilities can solve new offline RL problems in context: they infer near-optimal policies from a small offline dataset consisting of transitions in the unseen environment collected by suboptimal behavioral policies.

**Large Language Models and Autoregressive Decision Making.** Large Language Models and autoregressive models (Radford et al., 2019; Brown et al., 2020b; Wu et al., 2023b; Touvron et al., 2023; OpenAI et al., 2024) have achieved astonishing empirical successes in a wide range of application areas, including medicine (Singhal et al., 2023; Thirunavukarasu et al., 2023), education (Kasneci et al., 2023), finance (Wu et al., 2023a; Yang et al., 2023), etc. As it is natural to use autoregressive models for sequential decision making, transformer models have demonstrated superior performance in both bandit and MDP problems (Li et al., 2023; Yuan et al., 2023). In particular, Decision Transformer (DT) (Chen et al., 2021; Zheng et al., 2022; Liu et al., 2023; Yamagata et al., 2023) uses return-conditioned supervised learning to tackle offline RL. Although salable to multi-task settings (i.e., one model for multiple RL problems), DT is commonly criticised for its inability to improve upon the offline datasets and provably sub-optimal in certain scenarios, e.g., environment with high stochasticity (Brandfonbrener et al., 2022; Yang et al., 2022; Yamagata et al., 2023). More importantly, DT cannot generalize to unseen RL problems in context. To this end, Algorithm Distillation (AD) (Laskin et al., 2022) uses sequential modeling to emulate the learning process of RL algorithms, i.e., meta-learning (Vilalta & Drissi, 2002). The work most closely related to ours is the Decision Pretrained Transformer (DPT) (Lee et al., 2024), a supervised pretraining approach for in-context decision making. DPT trains transformers to predict the optimal action given a query state and a set of transitions. As delineated in Section 1, AD and DPT have stringent assumptions on the pretraining datasets. Our work overcomes those drawbacks and does not require query to optimal policies nor the complete learning histories of RL algorithms (Laskin et al., 2022; Lee et al., 2024).

## 3. Preliminary

**Markov Decision Process.** Sequential decision problems can be formulated as Markov Decision Processes (MDPs). An MDP is described by the tuple $\tau = (\mathcal{S}, \mathcal{A}, P, R, H, \rho)$ where $\mathcal{S}$ is the set of all possible states, $\mathcal{A}$ is the set of all possible actions, $P : \mathcal{S} \times \mathcal{A} \to \Delta(\mathcal{S})$ is the dynamic function that describes the distribution of the next state given the current state and action, $R : \mathcal{S} \times \mathcal{A} \to \Delta(R)$ is the reward function, $H \in \mathbb{N}$ is the horizon, and $\rho \in \Delta(\mathcal{S})$ is the initial state distribution. An agent (decision maker) interacts with the environment as follows. At the initial step $h = 0$, an initial state $s_0 \in \mathcal{S}$ is sampled according to distribution $\rho$. For all $h$ such that $0 \leq h \leq H$, the agent chooses action $a_h \in \mathcal{A}$ and receives reward $r_h \sim R(s_h, a_h)$. Then the next state $s_{h+1}$ is generated following the dynamic $P(s_h, a_h)$. A policy $\pi : \mathcal{S} \to \Delta(\mathcal{A})$ maps the current state to an action distribution. Let $V_\tau(\pi) = \mathbb{E}_\pi[\sum_{h=1}^{H} r_h]$ denote the expected cumulative reward of $\pi$ in $\tau$. The goal of an agent is to learn the optimal policy $\pi_\tau^*$ that maximizes $V_\tau(\pi)$.

**Decision-Pretrained Transformer.** DPT is a supervised pretraining method for transformers to have in-context RL capability. DPT assumes a set of tasks $\{\tau^i\}_{i=1}^m$ sampled from a task distribution $\mathcal{T}$. Here each $\tau^i$ is an instance of MDP. For each task $\tau^i$, a context dataset $D^i$ is next sampled, consisting of interactions between a behavioral policy and $\tau^i$, i.e., $D^i = \{(s_h^i, a_h^i, s_{h+1}^i, r_h^i)\}_h$, where $a_h^i$ is chosen by a behavioral policy. Additionally, for each task $\tau^i$, a query state $s_{\text{query}}^i \in \mathcal{S}$ is sampled, and an associated label $a_i^*$ is sampled from $\pi_{\tau^i}^*(s_{\text{query}})$, where $\pi_{\tau^i}^*$ is the optimal policy for $\tau^i$. The complete pretraining dataset is $D_{pre} = \{D^i, s_{\text{query}}^i, a_i^*\}_{i=1}^m$. Let $D^{i,j} = \{(s_h^i, a_h^i, s_{h+1}^i, r_h^i)\}_{h \leq j}$ denote the partial dataset of $D^i$ up to time step $j$. Let $T_\theta$ denote a causal GPT-2 transformer with parameters $\theta$ (Radford et al., 2019). The pretraining objective of DPT is defined as

$$\min_\theta \frac{1}{mH} \sum_{i=1}^m \sum_{h=1}^H - \log T_\theta \left( a_i^* | s_{\text{query}}^i, D^{i,h} \right). \quad (1)$$

**In-Context RL.** After pretraining, the pretrained DPT model $T_\theta$ can be deployed as both an online and offline agent. During deployment (testing), a test task $\tau$ is sampled from the testing task distribution $\mathcal{T}'$. Note that here the testing distribution $\mathcal{T}'$ can be different from the training distribution $\mathcal{T}$. For offline deployment, a dataset $D_{\text{off}}$ is first sampled from $\tau$, then DPT follows the policy $T_\theta(\cdot | s_h, D_{\text{off}})$ after observing the state $s_h$ at time step $h$. For online deployment, DPT initiate with an empty dataset $D_{\text{on}}$. In each episode, DPT follows the policy $T_\theta(\cdot | s_h, D_{\text{on}})$ to collect a trajectory $\{s_1, a_1, r_1, \ldots, s_H, a_H, r_H\}$ which will be appended into $D_{\text{on}}$. This process repeats for a pre-defined number of episodes. See Algorithm 3 (in Appendix) for detailed pseudocodes and Figure 1 for a visual demonstration.

# 4. Decision Importance Transformer

We present the *Decision Importance Transformer* (DIT), a pretraining approach that enables LLMs for in-context RL without requiring optimal action labels. As discussed in Section 1, given an offline dataset $D^i$ consisting of transitions $\{(s_h^i, a_h^i, s_{h+1}^i, r_h^i)_h\}$ collected by a suboptimal behavioral policy in the RL instance $\tau^i$, we aim to use the observed state-action pairs in $D^i$ as query states and pseudo-optimal action labels. The challenge here is to identify the *important* pairs where the actions are near-optimal for the corresponding states. To this end, DIT will assign a value $C_{\text{opt}}^i(s_h^i, a_h^i)$ to each state-action pair $(s_h^i, a_h^i)$ in the dataset, representing the optimality of action $a_h^i$ given $s_h^i$. A large value indicates that $a_h^i$ is a good action. Intuitively, if $a_h^i$ is a good action, then the agent should collect high rewards after time step $h$. Thus, we propose to define the values as follows:

$$C_{\text{opt}}^i(s_h^i, a_h^i) = \sum_{h'=h}^{H} \gamma^{h'-h} r_{h'}^i, \tag{2}$$

where $0 < \gamma < 1$ is a discounting factor. The purpose of $\gamma$ is to encourage $C_{\text{opt}}^i(s_h^i, a_h^i)$ to focus more on the immediate rewards, as future rewards are influenced not only by $a_h^i$ but also, and perhaps more significantly, by the subsequent actions $\{a_{h'}^i\}_{h'>h}$. While our proposed approach constructs values in a trajectory-independent manner, it is feasible to employ supervised in-context learning for values constructed using cross-trajectory information, as discussed in Appendix B. However, since the primary goal of this work is to demonstrate the effectiveness of weighted pretraining and in-trajectory pseudo-optimal labels, we opted for the simpler construction in (2).

When a pair $(s_h^i, a_h^i)$ is used as the query state and the pseudo-optimal action label for pretraining, if $a_h^i$ is near-optimal, i.e., $C_{\text{opt}}^i(s_h^i, a_h^i)$ is high, we should assign it more weight in the DPT pretraining objective in (1) because it is more likely to be the true optimal action. Hence, we propose the following Weighted Maximum Likelihood Estimation (WMLE) loss as the supervised pretraining objective of DIT:

$$\min_{\theta} L(\theta) = \frac{1}{mH^2} \sum_{i=1}^{m} \sum_{h'=1}^{H} \mathcal{M}\left(C_{\text{opt}}^i\left(s_{h'}^i, a_{h'}^i\right)\right) \mathcal{L}^i(\theta)$$
$$\text{with } \mathcal{L}^i(\theta) = \sum_{h=1}^{H} -\log T_{\theta}\left(a_{h'}^i | s_{h'}^i, D^{i,h}\right), \tag{3}$$

where $\mathcal{M} : \mathbb{R} \to \mathbb{R}$ is a monotone function for rescaling the assigned values. Intuitively, $\mathcal{M}$ serves to control the relative weight differences among the actions with large assigned values and those with small ones. For example, consider $\mathcal{M}(x) = \exp(\kappa x)$ where $\kappa \geq 0$. As $\kappa$ increases, actions assigned larger values will receive significantly more weight during pretraining. Compared with the pretraining objec-

tive of DPT in (1), DIT uses all observed state-action pairs $(s_h^i, a_h^i)$ in the offline dataset as query states and optimal action labels. For observed actions that are more likely to be good actions, DIT increases their associated weights in the pretraining objective. These weights effectively serve as the importance sampling ratio between the optimal and offline trajectories, guiding the suboptimal policy toward the optimal policy. We present pseudocode for DIT in Algorithm 1.

---

**Algorithm 1** Decision Importance Transformer for Markov Decision Processes

1: **Input:** Pretraining Dataset $D = \{D^i\}, D^i = \{(s_h^i, a_h^i, s_{h+1}^i, r_h^i)\}_h$; Scale Function $\mathcal{M}$; Discount Factor $\gamma$; Transformer Model $T_{\theta}$.
2: `// Calculating Importance Weights`
3: **for** each context dataset $D^i$ **do**
4:     **for** each state action pair $(s_h^i, a_h^i)$ **do**
5:         $C_{\text{opt}}^i(s_h^i, a_h^i) = \sum_{h'=h}^{H} \gamma^{h'-h} r_{h'}$
6:     **end for**
7: **end for**
8: `// Weighted MLE`
9: Randomly initialize $T_{\theta}$.
10: **while** not converged **do**
11:     Sample an context dataset $D^i$ from $D$ and a step index $j$. Compute the loss:

$$-\frac{1}{H} \sum_{h=1}^{H} \mathcal{M}\left(C_{\text{opt}}^i\left(s_j^i, a_j^i\right)\right) \log T_{\theta}\left(a_j^i | s_j^i, D^{i,h}\right)$$

12:     Backpropagate to update $\theta$
13: **end while**

---

**DIT for Bandit Problems**

Here, we provide a separate treatment for the bandit problems, which are special cases of MDP problems where the state set $\mathcal{S}$ contains only a single member, and the horizon of each episode is 1. For each action (bandit) $a \in \mathcal{A}$, there is an associated random reward $r(a)$. The optimal action is defined as $a^* \in \text{argmax}_a \mathbb{E}[r(a)]$.

The $i$-th context dataset $D^i$ for the bandit problems is a trajectory $\{a_1^i, r_1^i, \ldots, a_H^i, r_H^i\}$ or as a set of transitions $\{(a_h^i, r_h^i)\}_{h=1}^{H}$ with $r_h^i \sim r^i(a_h^i)$ where $r^i(a)$ denotes the random reward associated with bandit $a$ in the $i$-th bandit problem. With knowledge of the optimal bandits, the pretraining objective of DPT for bandit problems is

$$\min_{\theta} \frac{1}{mH} \sum_{i=1}^{m} \sum_{h=1}^{H} -\log T_{\theta}\left(a_i^* | 1, D^{i,h}\right), \tag{4}$$

where without loss of generality the single member of $\mathcal{S}$ is set to 1 (i.e., $\mathcal{S} = \{1\}$) and $a_i^*$ is the optimal action for the $i$-th bandit problem. After pretraining, DPT models can

generalize to unseen bandit instances, demonstrating performance matches that of the theoretically optimal algorithms for bandit problems (Lee et al., 2024). However, without information about the optimal bandits across different problems, this approach is not directly applicable. To this end, given a trajectory $\{a_1^i, r_1^i, \ldots, a_H^i, r_H^i\}$ without the optimal bandit $a_i^*$, DIT assigns a value $C_{\text{opt}}^i(a)$ to each $a \in \mathcal{A}$, that represents the optimality of $a$. A larger value of $C_{\text{opt}}^i(a)$ indicates that $a$ is a better action for the $i$-th bandit problem. A natural choice for $C_{\text{opt}}^i(a)$ is the expected reward of $a$, i.e., $C_{\text{opt}}^i(a) = \mathbb{E}[r^i(a)]$. In this case, the bandit with the highest $C_{\text{opt}}^i(a)$ is the optimal action. DIT first estimates $C_{\text{opt}}^i(a)$ with the context dataset: for any $a \in \mathcal{A}$,

$$\widehat{C_{\text{opt}}^i}(a) = \frac{\sum_{h=1}^H r_h^i \mathbb{1}\{a_h^i = a\}}{\sum_{h=1}^H \mathbb{1}\{a_h^i = a\}}. \tag{5}$$

Next, DIT creates a pseudo-optimal bandit $\widehat{a}_i$ for the $i$-th bandit problem with

$$\widehat{a}_i = \underset{a \in \mathcal{A}}{\operatorname{argmax}} \widehat{C_{\text{opt}}^i}(a).$$

For any bandit that is unobserved in the context dataset $D^i$, we assign a very small value to it, ensuring it will not be selected. With $\widehat{a}_i$'s as the estimations for the optimal bandits, DIT employs the following pretraining objective:

$$\min_{\theta} \frac{1}{mH} \sum_{i=1}^m \sum_{h=1}^H -\mathcal{M}\left(\widehat{C_{\text{opt}}^i}(\widehat{a}_i)\right) \cdot \log T_\theta\left(\widehat{a}_i | 1, D^{i,h}\right), \tag{6}$$

where $\mathcal{M} : \mathbb{R} \to \mathbb{R}$ is a monotone function for rescaling the assigned values. Here, pseudo-optimal bandits that have received more rewards (with larger $\widehat{C_{\text{opt}}^i}$) will be assigned greater weights during pretraining, as they are more likely to be the optimal bandits. We present pseudocode for the proposed algorithm in Algorithm 2.

# 5. Empirical Studies

We empirically demonstrate the efficacy of DIT through experiments on various bandit and MDP problems. In bandit problem, DIT showcases matching performance to that of the theoretically optimal bandit algorithms in both online and offline settings. In MDP problems, we corroborate that DIT can infer close-to-optimal policies from suboptimal pretraining datasets. Notably, albeit without optimal action labels during pretraining, DIT models demonstrate performance as strong as that of DPT models, which have access to optimal action labels during pretraining.

**Implementation.** To improve the stability and efficiency of training, we normalize the assigned values so that, after normalization, all values are positive (see Appendix D.1 for details). We set the discounting factor $\gamma$ to 0.8. After normalization, we apply the weight-scaling function $\mathcal{M}(x) = \lambda x^2$.

---

**Algorithm 2** Decision Importance Transformer for Bandit Problems

1: **Input:** Dataset $D = \{D^i\}$, $D^i = \{a_1^i, \ldots, r_H^i\}$; Scale Function $\mathcal{M}$; Transformer Model $T_\theta$.
2: **for** each context dataset $D^i$ and each bandit $a \in \mathcal{A}$ **do**
3: $\quad \widehat{C_{\text{opt}}^i}(a) = \frac{\left(\sum_{h=1}^H r_h^i \mathbb{1}\{a_h^i = a\}\right)}{\left(\sum_{h=1}^H \mathbb{1}\{a_h^i = a\}\right)}$
4: **end for**
5: Randomly initialize $T_\theta$.
6: **while** not converged **do**
7: $\quad$ Sample an context dataset $D^i$ from $D$.
8: $\quad$ Build the pseudo-optimal bandit $\widehat{a}_i = \operatorname{argmax}_{a \in \mathcal{A}} \widehat{C_{\text{opt}}^i}(a)$. Compute the loss:

$$\frac{1}{H} \sum_{h=1}^H -\mathcal{M}\left(\widehat{C_{\text{opt}}^i}(\widehat{a}_i)\right) \cdot \log T_\theta\left(\widehat{a}_i | 1, D^{i,h}\right)$$

9: $\quad$ Backpropagate to update $\theta$
10: **end while**

---

The choice of $\lambda$ is critical: it should be larger for offline datasets where optimal actions are observed less frequently so that the important yet rare good actions receive more weights during pretraining. In this work, we use a simple scheme with the same $\lambda$ value for all problems, setting $\lambda$ to 500 for all bandit and MDP problems. However, there is potential for improved weight-scaling using more complex schemes, such as varying values for different pretraining tasks.

## 5.1. Bandit Problems

We consider linear bandit (LB) problems with an underlying structure shared among tasks. Specifically, there exists a bandit feature function $\phi : \mathcal{A} \to \mathbb{R}^d$ that is *fixed* across tasks where $d$ denotes the dimension of the problems. The reward of a bandit $a \in \mathcal{A}$ in task $\tau^i$ is $r^i(a) \sim \mathcal{N}\left(\mu_a^i, \sigma^2\right)$ where $\mu_a^i = \mathbb{E}[r|a, \tau^i] = \langle \theta^i, \phi(a) \rangle$ and $\sigma^2 = 0.3$. Here, $\theta^i$ is the task specific parameter that defines task $\tau^i$. We conduct experiments on LB problems where $K = 20$ (total number of bandits), $d = 10$ and $H = 200$. The pretraining dataset for DIT are generated as follows.

**Pretraining Dataset.** For LB problems, we generate the feature function $\phi : \mathcal{A} \to \mathbb{R}^d$ by sampling bandit features from independent Gaussian distributions, i.e., $\phi(a) \sim \mathcal{N}_d\left(0, I_d/d\right)$ for all $a \in \mathcal{A}$. To generate the pretraining tasks $\{\tau^i\}$, we sample their parameters $\{\theta^i\}$ independently following $\theta^i \sim \mathcal{N}_d\left(0, I_d/d\right)$. To generate context dataset $D^i$, we randomly generate a behavioral policy by mixing (i) a probability distribution samples a Dirichlet distribution and (ii) a point-mass distribution on one random arm. The mixing weights are uniform sampled from $[0.0, 0.1, \ldots, 1.0]$.

At every time step $h$, the behavioral policy samples an action $a_h^i$ and receives $r_h^i$. We do not enforce extra coverage of the optimal actions for bandit problems. Following the setting of DPT (Lee et al., 2024), we collect 100k context datasets for LB problems.

**Comparisons.** We compare to the following baselines (see Appendix C for more details ): Empirical Mean (EMP) selects the bandit with the highest average reward; Upper Confidence Bound (UCB) (Auer, 2002) builds upper confidence bounds for all bandits and selects the bandit with the highest upper bound; Lower Confidence Bound (LCB) (Xiao et al., 2021) builds lower confidence bounds for all bandits and selects the bandit with the highest lower bound; Thompson Sampling (TS) (Russo et al., 2018) builds a posterior distribution for the rewards of all bandits. At each step, TS samples means for all bandits from the posterior distribution and selects the bandit with the highest sampled mean. In term of metrics, for offline learning we follow the convention to use the *suboptimality* defined as $(\mu_{a^*} - \mu_{\hat{a}})$ where $\mu_{a^*}$ is the mean reward of the optimal bandit and $\mu_{\hat{a}}$ is the mean reward of the chosen bandit; for online learning we use the *cumulative regret* defined as $\sum_h (\mu_{a^*} - \mu_{a_h})$ where $a_h$ is the chosen action at time step $h$.

**Empirical Results.** As can be seen in Figure 2, in the online setting, though pretrained without the optimal action labels, DIT models demonstrate superior performance to those of the theoretically optimal bandit algorithms, i.e., UCB and TS. Deployed for unseen bandit problems, DIT models quickly identify the optimal bandits at the beginning and maintain sub-linear regrets over the horizon. In the offline setting, DIT models can infer near-optimal bandits from trajectories collected by sub-optimal policies. In particular, when the behavioral policies (captioned 'BEH' in Figure 2) are randomly generated policies, DIT significantly outperforms both TS and LCB, the theoretically optimal algorithm for offline bandit problems. When the context is collected by expert policies, DIT models improve upon their performance, achieving lower regrets through in-context decisions.

### 5.2. MDP Problems

We conduct experiments on two MDP environments widely used by works about in-context learning: Dark Room (Laskin et al., 2022) and Miniworld (Chevalier-Boisvert et al., 2023). In Dark Room, the agent is randomly placed in a room of $10 \times 10$ grids, and there is an *unknown* goal location on one of the grid. The agent needs to move to the goal location by choosing from 5 actions: to move in one of the 4 directions (up, down, left, right) or stay still. The agent receives a reward of 1 only when it is at the goal; otherwise, it receives 0. The horizon for Dark Room is 100. In Miniworld, the agent is placed in a room with four boxes

of different colors, one of which being the target box. The agent receives a $(25 \times 25 \times 3)$ color image and its direction as input, and can choose from four possible actions: to turn left/right, move straight forward, or stay still. Similar to Dark Room, it receives a reward of 1 only when it is near the target box while the horizon is 50. The optimal policies for both MDP problems are known exactly. See Appendix C.3 for details of these environments.

**Pretraining Datasets.** To ensure coverage of optimal actions (so that optimal policies can be inferred), at every step, with probability $p$ (respectively $1 - p$) we use optimal policy (respectively random policy) to choose action. We choose $p$ so that the average reward of the trajectories in the pretraining dataset is less than $30\%$ of that of the optimal trajectories, reflecting the challenging yet common scenarios. For Dark Room, to test whether DIT models can generalize to unseen RL problems in context, we collect context datasets from only 80 out of the total 100 goals and reserves the rest 20 for testing. For each training goal, we follow the setting of DPT to collect 1k context datasets, leading to a total of 80k context datasets in the pretraining dataset (64k for training and 16k for validation). For Miniworld, we collect 40k context datasets (32k for training and 8k for validation), 10k datasets for each of the four tasks corresponding to four possible box colors.

**Comparisons.** We compare DIT to other in-context algorithms as well as RL algorithms that train an agent from scratch without pretraining. The baseline algorithms are briefly described next (see their implementation details in Appendix C).

- Proximal Policy Optimization (PPO) (Schulman et al., 2017): PPO is an online algorithm that trains an agent from scratch in every environment.

- Algorithm Distillation (AD) (Laskin et al., 2022): AD is an in-context algorithm whose pretraining dataset consists of learning histories of an RL algorithm in the pretraining tasks. In this work we use PPO as the RL algorithm for AD. AD can be both online and offline algorithms.

- Decision Pretrained Transformer (DPT): DPT and DIT use the same context datasets for pretraining. However, DPT requires query states across different tasks and their associated optimal action labels. We follow the original setting of DPT to uniformly sample query state from all possible states and obtain an associated optimal action label.

- Behavioral Clone (BC) (Hussein et al., 2017): BC imitates the behavioral policies of the pretraining dataset

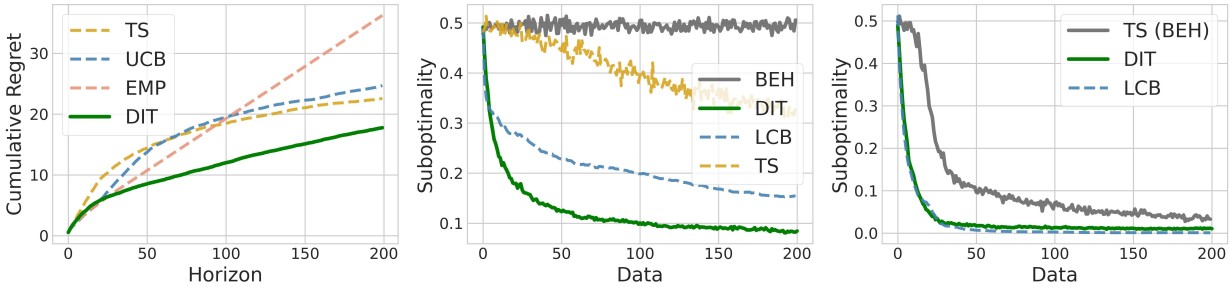

*Figure 2.* Results for Linear Bandits (lower values indicate better performance). **Left**: Online testing. **Middle**: Offline testing conditioned on trajectories gathered by highly suboptimal, randomly generated policies. **Right:** Offline testing condtioned on trajectories gathered by experts (Thompson Sampling policies).

by optimizing

$$\min_\theta \frac{1}{mH} \sum_{i=1}^{m} \sum_{h=1}^{H} - \log T_\theta \left( a_h^i | s_h^i, D^{i,h} \right).$$

- Weighted Behavioral Clone (WBC): WBC is the weighted-version of BC. During pretraining, it optimizes

$$\min_\theta \frac{1}{mH} \sum_{i=1}^{m} \sum_{h=1}^{H} -C_{i,h} \log T_\theta \left( a_h^i | s_h^i, D^{i,h} \right)$$

where $C_{i,h} = \mathcal{M}(C_{\text{opt}}^i(s_h^i, a_h^i))$ is the same weight used by DIT. This method functions as a baseline to evaluate the effectiveness of the weight-by-optimality principle employed by DIT.

In terms of metrics, we follow the convention to use the episode cumulative return $\sum_{h=1}^{H} r_h$.

**In-Context Decision for Unseen Tasks.** We explore how our method generalizes to unseen RL tasks, using the Dark Room environment (Laskin et al., 2022). Following the approach of DPT (Lee et al., 2024), we use 80 goals for training and evaluate on the remaining 20 unseen goals. For PPO (Schulman et al., 2017), since it is an online learning method, we directly train from scratch on the 20 goals to benchmark the returns of in-context RL. Figure 3(a) shows the online evaluation over 40 episodes. After 40 episodes, PPO gains little in return, demonstrating the difficulty of the RL tasks for testing. Restricted by their capability to efficiently explore in new tasks, imitation learning methods (i.e., BC and WBC) also perform poorly. Although our method (DIT) initially has lower returns than DPT and AD, it quickly surpasses them and continues to improve. Figures 3(b) and 3(c) show the results for offline evaluations with expert (high-reward) trajectories and random (low-reward) trajectories. Despite being pretrained *without* the optimal action labels, DIT models demonstrate competitive performance to that of DPT. Notably, WBC outperforms

BC in both scenarios, further indicating the effectiveness of the proposed weighted-pretraining framework.

**Comparison with Oracles.** In this section, we conduct experiments in the Miniworld (Chevalier-Boisvert et al., 2023) environment to investigate whether DIT reaches the limits of the pretraining datasets. To this end, we compare our model to the DPT model that uses a pretraining dataset containing only *query states that belong to the set of observed states in the pretraining dataset*, along with their associated optimal action labels. In this scenario, the total number of pretraining context datasets and optimal action labels remains the same, but the query states are restricted. This restriction makes the DPT model function as an oracle upper bound for DIT, as all query states for DIT originate from the observed states. It is worth noting that AD is trained with a different pretraining dataset that has more stringent assumptions (i.e., learning histories), while the other methods use the same pretraining dataset, except that DPT also employs optimal action labels. Surprisingly, in the online setting, DPT struggles to perform, while DIT models gradually improve their returns, as shown in Figure 4(a). In the offline setting, DIT again demonstrates competitive performance with DPT, indicating that it has effectively leveraged the pretraining dataset to a significant extent.

**Ablation Studies.** Here we investigate the effect of the weights in WMLE loss during pretraining. We pretrain DIT models on two pretraining datasets for Dark Room, one with high average reward (30% average reward of the optimal policies) and the other with low reward (10% of that of the optimal policies). For each pretraining dataset, we train two DIT models respectively with small and large weights for the WMLE loss. Hence, we have 4 DIT models in total. As can be seen from Figure 5, the scales of the weights are crucial. During online testing, the model pretrained on the low-reward dataset with a small weight cannot fully utilize the dataset. Increasing the weight significantly improves its performance (i.e., 'L:2x' outperforms 'L:1x'). In contrast, when pretraining on a high-reward dataset, using a too large

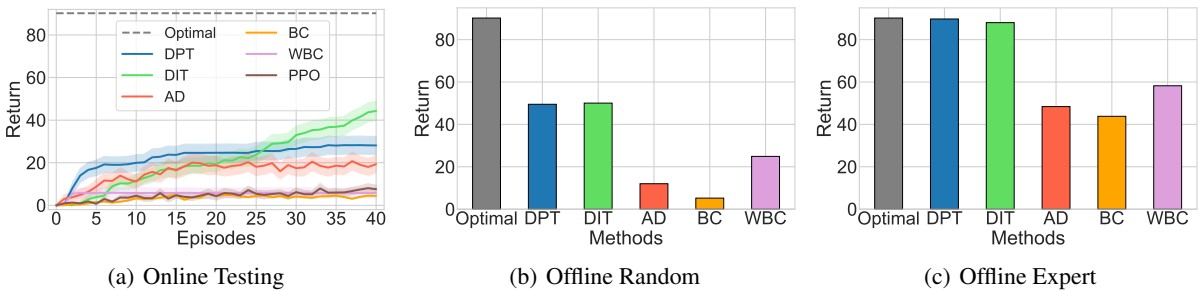

*Figure 3.* Results on Dark Room (higher values indicate better performance). **Left:** change in return of policies with additional online episodes for (in-context) learning. **Middle** and **Right:** offline evaluations with context trajectories sampled from random and expert policies.

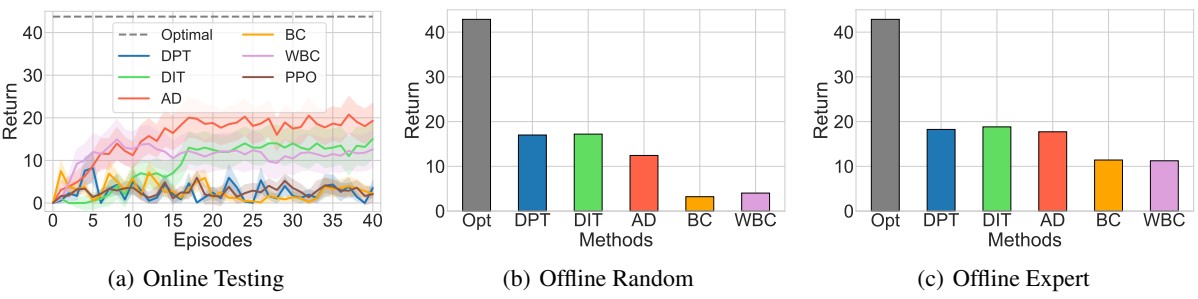

*Figure 4.* Comparison of In-context RL Algorithms on Miniworld.

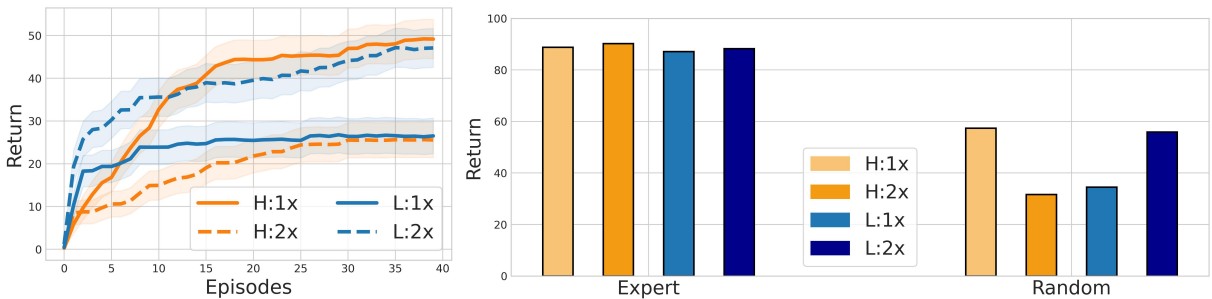

*Figure 5.* Models pretrained with high-reward (low-reward) datasets are labeled as 'H' ('L'); models pretrained with large weights (small weights) for the WMLE loss are labeled as '2x' ('1x'). **Left**: Online testing. **Right**: Offline testing with expert and random trajectories.

weight degrades performance. This empirically corroborates that, given a pretraining dataset with lower rewards, the rarer good actions should receive more weights during supervised pretraining. We observe the same behavior for offline testing with low-reward random trajectories. The performance for offline testing with expert trajectories shows little variation. We believe this is due to the simplicity of offline in-context RL with optimal trajectories.

## 6. Discussion

We have proposed DIT for pretraining LLMs without optimal action labels. DIT has demonstrated superior empirical performance: it matches the performance of theoretically optimal bandit algorithms and can infer near-optimal decisions with highly sub-optimal offline datasets. Despite these strengths, DIT still requires the optimal actions to be observable in the pretraining dataset because it is unlikely, if not entirely unfeasible, to infer optimal actions all from random trajectories and without any information about optimal policies. The appropriate weight scale for the weighted MLE loss during pretraining is also an important question to investigate. While we have established qualitative understandings through experimental results, there should exist a theoretically optimal scale based on the quality of the pretraining dataset. We will pursue this in future work.

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

## A. Pseudocodes

---

**Algorithm 3** Deployment of In-Context RL Models

---

**Input:** Pretrained Transformer Model $T_\theta$; Horizon of episodes $H$; Number of episodes $N$ for online testing; Offline dataset $D_{\text{off}} = \{(s_h, a_h, s_{h+1}, r_h)\}_h$, consisting of transitions collected by a behavioral policy. // `Offline Testing` every time step $h \in \{1, \dots, H\}$ Observe state $s_h$ Sample action with $T_\theta$:

$$a_h \sim T_\theta \left( \cdot | s_h, D_{\text{off}} \right)$$

Collect reward $r_h$ // `Online Testing` Initialize an empty online data buffer $D_{\text{on}} = \{\}$ every online trial $n \in \{1, \dots, N\}$ every time step $h \in \{1, \dots, H\}$ Observe state $s_h$ Sample action with $T_\theta$:

$$a_h \sim T_\theta \left( \cdot | s_h, D_{\text{on}} \right)$$

Collect reward $r_h$ Append the collected transitions $\{(s_h, a_h, s_{h+1}, r_h)\}_h$ into $D_{\text{on}}$

---

## B. In-Context Learning for Weights in Pretraining

Supervised ICL (Akyürek et al., 2022; Dong et al., 2022; Min et al., 2022) addresses the problem where, given a set of datasets from diverse supervised learning tasks $\{D^i\}_i$ where $D^i = \{(x_j^i, y_j^i)\}_j$ is generated by an underlying truth function $g^i : \mathcal{X} \to \mathcal{Y}$ as $y_j^i = g^i(x_j^i) + \epsilon$ with $\epsilon$ as a zero-mean noise. Here all the underlying functions $\{g^i\}$ belong to a task function class $\mathcal{F}$ with structures, e.g., linear functions. Supervised ICL employs a transformer model to interpolate across the given datasets by optimizing the objective

$$\min_\theta L_{\text{ICL}}(\theta) = \frac{1}{m} \sum_{i=1}^{m} \sum_{j=1}^{n_i} \ell(T_\theta(D^i(-j), x_j), y_j), \tag{7}$$

where $\ell : \mathcal{Y} \times \mathcal{Y} \to \mathbb{R}$ is the loss function depending on the application (e.g., mean-squared-error for regression) and $D^i(-j)$ is the dataset identical to $D^i$ but with $(x_j^i, y_j^i)$ removed.

The transformer models trained with (7) as their objective functions demonstrate strong performance on both instances from the training tasks as well as those from unseen tasks in $\mathcal{F}$. By interpolating across training tasks and extracting their shared structure, supervised ICL can solve the given training tasks with only a small batch of training samples for each task. Of course, it requires a significant number of training tasks so that the shared structure across tasks can be inferred.

In our work, the values assigned to observed state-action pairs can also be considered as a supervised ICL task. In particular, the input is $(s_h^i, a_h^i)$ and the output is the value to assign, with $C_{\text{opt}}^i$ defined in (2) functioning as noisy label for regression. This approach has the potential to interpolate across tasks, leading to more consistent value assignments over tasks. Yet, since the primary goal of this work is to demonstrate the effectiveness of weighted pretraining and in-trajectory pseudo-optimal labels, we opted for the simpler construction in (2).

## C. Baselines

### C.1. Bandit Algorithms

**Empirical Mean (EMP).** We follow (Lee et al., 2024) to consider a strengthened version of EMP which, in the offline setting, only chooses from actions that have been observed at least once in the offline dataset while, in the online setting, at least choosing every action once. At every time step, EMP chooses actions as

$$\widehat{a} \in \underset{a \in \mathcal{A}}{\text{argmax}} \{\widehat{\mu}_a\},$$

where $\widehat{\mu}_a$ is the average observed reward for action $a$.

**Upper Confidence Bound (UCB).** Motivated by the Hoeffding's Inequality, at each time step, UCB chooses actions as

$$\widehat{a} \in \underset{a \in \mathcal{A}}{\text{argmax}} \left\{ \widehat{\mu}_a + C \cdot \sqrt{1/n_a} \right\},$$

where $C$ is a hyperparameter and $n_a$ is the number of times $a$ has been chosen. For unseen actions, $\widehat{\mu}_a$ is set to 0 and $n_a$ is set to 1. We follow (Lee et al., 2024) to set $C$ to be 1 as it demonstrates the best empirical performance.

**Lower Confidence Bound (LCB).**   LCB is on the contrary of UCB. In the offline setting, LCB only chooses from observed actions in the offline dataset. Specifically, it chooses actions as

$$\widehat{a} \in \underset{a \in \mathcal{A}}{\operatorname{argmax}} \left\{ \widehat{\mu}_a - C \cdot \sqrt{1/n_a} \right\},$$

where $C$ is a hyperparameter and $n_a$ is the number of times $a$ has been chosen. Similar to hyperparameter of UCB, the hyperparameter $C$ for LCB is also set to 1 due to its strong empirical performance.

**Thompson Sampling (TS).**   We use Gaussian TS (Russo et al., 2018) with a Gaussian prior. The mean and variance of the prior are set to the true mean and variance of the pretraining tasks: 0 for mean and 1 for variance.

### C.2. RL Baselines

**Proximal Policy Optimization (PPO).**   We use the Stable Baselines3 implementation(Raffin et al., 2021) for Proximal Policy Optimization (PPO) and keep the default parameter settings. We train for 1,000 episodes per goal in the Dark Room environment and 2,500 episodes for Miniworld. For both environments, the policy model is implemented as a multi-layer perceptron with 2 layers of 64 units each. Additionally, we use a convolutional neural network with 2 convolutional layers, each with 16 kernels of size $3 \times 3$, followed by a linear layer with an output dimension of 8.

**Algorithm Distillation (AD).**   During PPO training, we collect the learning history for use in AD. According to the algorithm (Laskin et al., 2022), the model takes a cross-episodic trajectory $T$ of length $H$, representing the horizon. It is trained to predict the action taken after $K$ episodes following $T$. Here, we choose $K = 100$ as it helps to speed up the training. In both environments, we use the same transformer architecture as in DPT and DIT. For Miniworld, we use the same CNN architecture as in PPO works as the image encoder.

**Decision-Pretrained Transformer (DPT).**   The Decision-Pretrained Transformer (DPT) uses supervised learning to predict optimal actions. The model operates by considering the complete history of an episode and a specific query state sampled from the environment to predict the next action. This setup allows DPT to leverage past experiences as context for making decisions about future actions. To prepare training data for DPT, a complete episode is first generated by a predefined policy. A query state is then sampled from the environment, and the corresponding optimal action serves as the label. The model is trained to predict the optimal action given the episode as context based on the query state. The transformer for DPT is the same as in AD.

**Behavior Clone (BC) and Weighted Behavior Clone (WBC).**   In Behavior Clone (BC), given an episode, the model is trained to predict the next action based on the current state and the past trajectory. We sample episodes based on a policy that, with probability $p$, takes an action from the optimal policy, and with probability $1 - p$, takes an action from a random policy. We set $p = 0.2$ in the Dark Room and $p = 0.7$ in Miniworld. For Weighted Behavior Clone (WBC), when calculating the loss for each action taken from a state, we reweight it by the cumulative rewards from that action until the end of the episode. This ensures the model learns actions that have a higher influence on the rest of the episode.

**Training Parameters.**   For DIT, DPT, AD, BC, and WBC, we use the AdamW optimizer with a weight decay of 1e-4, a learning rate of 1e-3, and a batch size of 64.

### C.3. MDP Environments

**Dark Room.**   The agent's observation is its current position/grid in the room, i.e., $\mathcal{S} = [10] \times [10]$. The goal is in one of the grids. Thus, there are $10x10 = 100$ goals. It can choose from five actions: left, right, up, down, and stay. We follow (Lee et al., 2024) to use the tasks on 80 out of the 100 goals for pretraining, and reserve the rest 20 goals for testing our models' in-context RL capability for unseen tasks. The optimal actions are defined as: move up or down until the agent is on the same vertical position as the goal; otherwise move left or right until the agent reaches the goal.

**Miniworld.** In the Miniworld-Image, the agent's observation is a $(25 \times 25 \times 3)$ color image and a 2-D direction. The goal is to reach a box of a specific color in the room. In both cases, the agent can choose actions from: turn left, turn right, move forward, and stay. The optimal actions are defined as follows: turn left/right towards the correct box if the agent's front is not within 15 degrees of the correct box; otherwise move forward and stay if the agent is near the box.

## D. Experimental Details

### D.1. Value Normalization

For bandit problems, we normalize as follows: $\widehat{C_{\text{opt}}^i}(a) \leftarrow \max\left(\widehat{C_{\text{opt}}^i}(a) - B_{\text{bandit}}^i, 0\right)$ where $B_{\text{bandit}}^i$ the average assigned value across all actions in the $i$-th pretraining dataset. The $\max$ operation ensures that below-average actions are not used as pseudo-optimal action labels. We use a similar formula to normalize for MDP problems: $C_{\text{opt}}^i(s, a) \leftarrow \max\left(C_{\text{opt}}^i(s, a) - B_{\text{MDP}}^i\right)$ where $B_{\text{MDP}}^i = \sum_{h=1}^{H} C_{\text{opt}}^i(s_h^i, a_h^i)/H$ is the average assigned value across all state-action pairs.

### D.2. Model Details

**Transformer Models.** For bandit problems and experiments on Dark Room, we follow the recommended settings of (Lee et al., 2024) to use a causal GPT2 model (Radford et al., 2019) for DPT. The transformer models for DPT has an 4 attention layers each with 4 attention heads and an embedding size of 32. The DIT models use the same transformer models.

### D.3. Computation Requirements

Our experiments can be conducted on a single A6000 GPU. It typically takes less than one hour to generate the required dataset for training in parallel. For PPO, training usually takes less than 10 minutes per task. For the other methods, we observe that the transformer model converges within 50 epochs.

