# OpenReview forum: "In-Context Reinforcement Learning Without Optimal Action Labels"
_ICML.cc/2024/Workshop/ICL — ICML 2024 Workshop ICL Poster_

### Official Review · Reviewer_Y2Sh · 2024-06-06
**Review of IC RL without Optimal Action Labels**

**Rating:** 2
**Fit:** 3
**Confidence:** 3

**Workshop Review:**

The paper proposes a "Decision Importance Transformer" (DIT) for in-context reinforcement learning. The key innovation from the authors is a modification to the loss used to train Decision Pretrained Transformers to help address the previous problem of DPT always requiring optimal actions in the context in order for the transformer to output the optimal action for a query state.

While the authors approach is interesting, and does show some promise for helping resolve this problem, this work has a couple drawbacks that are worth mentioning. The first is that the authors approach is not very novel. This optimization is a very simple extension of the previous training procedure for DPT, where they add a re-weighting mechanism that identifies the "important" state-pairs by multiplying each pair by the future return (so that actions that lead to higher future reward are considered to be more likely optimal for that given state). Its worth noting that this additional re-weighting mechanism comes with an additional hyperparameter lambda that the authors state their approach is highly-sensitive to. This weighting mechanism seems like it is only effective if the immediate rewards in the near horizon are a good proxy for the optimality of an action (which in many cases is not true like in long horizon sparse reward tasks that require farsighted behavior).

Additionally, given this additional hyperparameter introduced, the results are mildly promising. For the bandit results presented, the authors claim their DIT outperforms the optimal bandit algorithms like UCB and Thompson Sampling etc. However, it is odd that DPT is not a baseline here. The authors don't provide an explanation for why DPT isn't included in the bandit section, even though it is later included in the MDP section. Additionally, it seems that the comparison between UCB and DIT are a bit confusing in the online testing. DIT already assumes it has some context provided to the transformer in the Dataset input. Are the samples in the dataset counted towards the "horizon" in the online setting? It seems like according to Algorithm 2, the DIT is already starting with H samples as the context, while UCB would be starting from scratch, which doesn't seem like a fair comparison. Even still, in the DPT paper, the authors did not find that DPT outperforms the state of the art bandit algorithms, and the authors do not explain why in their paper DIT outperforms these optimal algorithms. Lastly, there are no error bars and uncertainties reported for the bandit results.

In the MDP case, it appears that DIT only does very slightly better than DPT in the Offline Random case, and does slightly worse in the Offline Expert case. In essence, it seems to just recover the performance of DPT for the offline setting for dark world. For the Miniworld case, algorithm distillation is still the strongest and DIT does outperform DPT which is promising, but once again there is no difference in the offline setting.

That being said overall this paper is interesting and clear and well-written and takes a first step in helping address an important problem in in-context RL requiring optimal actions in the context.

**Reason For Not Giving Higher Score:**

For the workshop fit: N/A

For the Workshop rating: This work does propose an interesting idea to solve the important problem in in context RL where optimal actions are required in the context for a transformer to learn an optimal policy. However, their approach is a very simple extension of DPT which has already been proposed, and the empirical results are not extremely strong. In the bandit case, DPT is not a baseline. In the MDP case, DIT achieves the same level of performance as DPT, with the occasional very slight improvement in the offline setting. There is some improved performance for the online setting, although the primary motivation given by the authors for this work is to improve the offline setting with suboptimal behavior policies.

**Reason For Not Giving Lower Score:**

Workshop fit: This work proposes an improvement to the state of the art of in-context reinforcement learning, which I think aligns perfectly with the overall purpose of the workshop on in-context learning.

Workshop Rating: Despite some of the issues with the novelty mentioned above, the approach taken by the authors is interesting and well motivated by the importance sampling ratio, a common technique in RL. Even though their empirical results are not extremely strong, their work leaves room for improvement going forward as they mentioned by introducing some new methods for tuning their $\lambda$ hyperparameter.

---

### Official Review · Reviewer_wkcf · 2024-06-07
**Review of In-Context Reinforcement Learning Without Optimal Action Labels**

**Rating:** 3
**Fit:** 3
**Confidence:** 2

**Workshop Review:**

This paper proposes to supervised pre-train transformer models using suboptimal action labels. Unlike some prior work in in-context RL which assumes that optimal action labels are available in the offline pre-training dataset, this paper removes that assumption by changing the behavior cloning objective to a weighted one. The weight is related to the overall optimality of the action, as computed by the discounted sum of rewards. Empirically, it seems to recover the performance of DPT, which requires optimal actions, in problems like Dark Room and Miniworld. The paper is also written clearly and is highly relevant to this workshop and general community.

**Reason For Not Giving Higher Score:**

N/A

**Reason For Not Giving Lower Score:**

This paper has thorough experimental results to show that optimal action labels are not necessary for supervised pre-training of decision transformers.

---

### Meta-Review · Area_Chair_WBna · 2024-06-16

**Recommendation:** 2

**Metareview:**

The paper introduces an innovative approach to supervised pre-training of transformer models using suboptimal action labels, addressing a critical challenge in in-context reinforcement learning (RL) where optimal action labels are typically assumed. By implementing a weighted behavior cloning objective, the authors' method, the Decision Importance Transformer (DIT), successfully recovers the performance of Decision Pretrained Transformers (DPT) in tasks such as Dark Room and Miniworld, even with suboptimal actions. The paper is well-written, highly relevant, and presents thorough experimental results that support its claims. The approach is both interesting and promising, showing potential for significant impact in the field. However, to further strengthen the paper, it would be beneficial to include DPT as a baseline in all relevant experiments and provide a more detailed explanation of the performance differences observed. Overall, this work makes a valuable contribution to the field of in-context RL and is recommended for acceptance.

---

### Decision · Program_Chairs · 2024-06-17

Accept (Poster)